# Photodynamic Therapy, Photobiomodulation and Acetonide Triamcinolone 0.1% in the Treatment of Oral Lichen Planus: A Randomized Clinical Trial

**DOI:** 10.3390/pharmaceutics15010030

**Published:** 2022-12-22

**Authors:** Carmen Salinas-Gilabert, Francisco Gómez García, Fe Galera Molero, Eduardo Pons-Fuster, Seppe Vander Beken, Pia Lopez Jornet

**Affiliations:** 1Faculty of Medicine and Odontology, Biomedical Research Institute (IMIB-Arrixaca) Hospital Morales Meseguer, Clínica Odontológica, Marqués del los Vélez s/n, 30008 Murcia, Spain; 2Departamento de Anatomía Humana y Psicobiología, Faculty of Medicine and Odontology, Biomedical Research Institute (IMIB-Arrixaca), University of Murcia, 30100 Murcia, Spain; 3Bredent Medical GmbH & Co. KG, 89250 Senden, Germany

**Keywords:** oral lichen planus, photobiomodulation, photodynamic therapy, oral pathology, topical corticosteroids

## Abstract

Objective: To evaluate the efficacy of photodynamic therapy (PDT) and photobiomodulation (PBM) in the treatment of oral lichen planus (OLP) in comparison with the use of topical corticosteroids. Material and methods: Sixty patients with OLP were randomized to three groups: group 1 photodynamic therapy applied once a week for four sessions, with orabase cream; group 2 low-power laser application with orabase cream; and group 3 inactive laser with triamcinolone acetonide 0.1%. Patient pain was evaluated, and the Thongprasom severity score, the Oral Health Impact Profile-14 (OHIP-14), and the Hamilton anxiety and depression scale at one and three months of follow-up. (ClinicalTrials.gov Identifier: NCT05127083). Results: Pain decreased significantly over time in all groups, though the symptoms relapsed over follow-up at one and three months in group 3. The OHIP-14 score improved significantly in groups 1 and 2 (*p* < 0.05), and this improvement was maintained after three months. Lesion resolution evaluated by the Thongprasom score at one month showed significant differences between groups 1 and 3 (*p* = 0.032) and between groups 2 and 3 (*p* = 0.024). Conclusions: Photodynamic therapy and photobiomodulation once a week for four weeks are safe and non-invasive treatment options, with the important advantage of lacking adverse effects. Further studies are needed to confirm it.

## 1. Introduction

Oral lichen planus (OLP) is a chronic mucocutaneous inflammatory disease that manifests in the form of flare-ups or outbreaks and affects 0.2–1.9% of the general population [1,2]. Although the underlying etiology is unclear, the disease is known to be characterized by an immune disorder with CD8-positive cytotoxic lymphocyte attack upon the epithelium. Clinically, OLP manifests as papular and reticular lesions that usually alternate with areas of erythema and atrophy and exhibit a dynamic behavior [1,2].

At present, the management of OLP is mainly based on drug treatments [3,4]. Different therapeutic strategies have been used to alleviate the symptoms, including topical and systemic corticosteroids, retinoids, and calcineurin inhibitors. Most of the studies involving drug treatments report improvement over short follow-up periods. However, many patients experience a relapse of symptoms once the treatment is suspended, with a negative impact on quality of life [4,5,6]. In this scenario, treatment alternatives capable of improving the symptoms with minimum side effects are desirable. Such alternatives include photodynamic therapy (PDT) and photobiomodulation (PBM) [7,8,9,10]. Photodynamic therapy is safe and simple to apply. It involves the induction of cell and tissue damage by combining a photosensitizer with low-power visible light of an adequate wavelength that activates the former [10]. On the basis of the first published studies, PDT was used to treat actinic (solar) keratosis and different types of skin cancer, such as basal cell carcinoma. Likewise, over the last 15 years, it has been used as a treatment for OLP [11,12,13,14,15,16]. Photodynamic therapy is minimally invasive and has the advantage of being very selective—thereby constituting an option for the treatment of OLP [13,14,15,16,17,18,19,20,21,22]. On the other hand, PBM mainly acts by increasing the production of adenosine triphosphate (ATP) and causing a brief burst in the generation of reactive oxygen species (ROS) that exert a beneficial effect upon the inflammatory process [11,12]. The use of PBM under different inflammatory conditions has an analgesic potential, with biostimulating and immune-modulating effects, and moreover improves healing—with the advantage over the current treatments of not producing major side effects [8,10].

A number of studies have evaluated PDT and PBM in the management of OLP, with contradictory results versus the use of corticosteroids [8,9,10,11,12,13,14,15,16,17,18,19,20,21,22,23,24,25,26]. The potential role of PDT and PBM in application to OLP is, therefore, subject to controversy [8,10,20,21,22,23,24,25,26].

The present study was carried out to evaluate the efficacy of PDT and PBM in the treatment of OLP, compared with the use of topical corticosteroids in the form of triamcinolone acetonide 0.1%, as assessed by the patient clinical signs and symptoms, and quality of life.

## 2. Material and Methods

### 2.1. Study Population

A randomized prospective study was carried out at the Dental Clinic of the University of Murcia (Murcia, Spain) involving patients diagnosed with OLP according to the criteria of van der Mej and van der Wal [27]. Subjects under 18 years of age were excluded from the study, as were patients subjected to corticosteroid therapy in the two months prior to the study, pregnant or nursing women, and individuals with decompensated systemic disorders or the presence of dysplasia in the histopathological study.

The study was carried out in line with the recommendations of the Declaration of Helsinki and was approved by the Ethics Committee of the University of Murcia (ID: 2227/2018). Written informed consent was obtained from all the patients (ClinicalTrials.gov Identifier: NCT05127083).

### 2.2. Study Design

A randomized, prospective four-week study was carried out, following the specifications of the Consort Statement (http://www.consort-statement.org/) (Figure 1). Randomization was carried out by an external laboratory using https://www.randomizer.org. The patients were blinded to their group assignment. The tubes of orabase cream and triamcinolone acetonide 0.1% plus orabase were all opaque and identical, with no identifying marks, and were prepared by the same laboratory. The randomization code was kept in a sealed envelope and opened immediately before the interventions. All data were collected by a single investigator (CSG). The following groups were established:-Group 1: Patients with OLP subjected to photodynamic therapy (Helbo^®^ blue photosensitizer with low-power laser irradiation) applied once a week for 4 sessions, with orabase cream application three times a day at home.-Group 2: Patients with OLP subjected to low-power laser irradiation applied once a week for 4 sessions, with orabase cream application three times a day.-Group 3: Patients with OLP subjected to inactive laser application once a week for 4 sessions with triamcinolone acetonide orabase cream 0.1% applied three times a day.

### 2.3. Application of Laser Phototherapy

Photodynamic therapy was carried out in 4 sessions on days 1, 7, 14, and 28. The Helbo^®^ blue photosensitizer gel containing 1% (*w*/*w*) methylene blue was applied for three minutes to a previously dried mucosal OLP lesion. After thorough cleansing with sterile saline solution, the lesion was irradiated with a Helbo^®^ 2D probe tip during 30 s/spot (active surface area 19 mm^2^) using a low-power laser (Helbo^®^ Theralite Laser, Bredent^®^ Medical GmbH & Co. KG, Senden, Germany) Energy-density (fluence) = 30 s × 200 mW/cm^2^ = 6 J/cm^2^.

The same procedure was carried out in group 2, though without the application of the photosensitizer, while in group 3, the inactive laser was used in each session. The election of the study’s sample size follows the next premises: an alpha 0.05 (95% confidence) and a 0.2 beta (0.8 power) risk are accepted in a bilateral contrast for the comparison of means between three independent groups, assuming unknown but equal variances to determine an effect size of high magnitude (*η*^2^ = 0.14). The number of patients is estimated to be 20 per group. However, 20% more patients will be included in those groups in case of loss. So, the final sample size will be 60 patients (20 per group).

### 2.4. Study Variables

The patients underwent clinical exploration of the oral cavity to assess the OLP lesions and the inclusion criteria. The intensity of the oral symptoms (pain) was recorded based on a visual analog scale (VAS) from 0–10 (where 0 = no pain and 10 = extreme pain).

The severity of the lesions was assessed based on the Thongprasom score (6), ranging from 0–5 points:

5 points = white lines with erosive area > 1 cm^2^ with erosive area

4 points = white lines with atrophic area < 1 cm^2^ with erosive area

3 points = white lines with atrophic area > 1 cm^2^

2 points = white lines with atrophic area < 1 cm^2^

1 point = mild white lines without erythema

0 points = no lesion; normal mucosa

The Oral Health Impact Profile-14 (OHIP-14) was used to explore the patient’s perceived severity and frequency of oral problems in relation to physical, psychological, and social aspects during the last month. This instrument consists of 7 domains (with two items per domain): functional limitation, physical pain, psychological discomfort, physical disability, psychological disability, social disability, and incapacitation. The higher the score, the greater the perceived negative impact on oral health-related quality of life [28].

The self-administered 14-item Hamilton anxiety and depression scale (HAD) [29] was also applied. This questionnaire consists of two subscales of 7 items each, scored based on a 0–3 Likert scale. The uneven-numbered items correspond to the anxiety subscale (HAD-A), and the even-numbered items to the depression subscale (HAD-D), with a score range for each subscale from 0–21. Higher scores are indicative of greater anxiety and depression.

After assessing the inclusion and exclusion criteria, the following study protocol and timelines were established:-Visit 1 (day 1): Thongprasom + VAS + OHIP-14 + HAD + treatment according to group-Visits 2 (day 7), 3 (day 14), and 4 (day 21): VAS + laser + treatment according to group-Visit 5 (1 month after the last visit): follow-up period with Thongprasom + VAS + OHIP-14 + HAD-Visit 6 (3 months after the fourth visit): follow-up period with VAS + OHIP-14 + HAD

### 2.5. Statistical Analysis

The effects of the within-subject factors (measures obtained at the visits) and between-subject factors (treatment) and their interactions (treatment * visit) upon the dependent variables (constants and scales) were analyzed using two-way repeated measures analysis of variance (ANOVA) verifying in each group whether variances were homogeneous. The differences in means between the visits were analyzed in each of the treatment modalities, along with the *p*-values of the two-by-two or post hoc comparisons made (Bonferroni correction). The statistical analysis was performed using the SPSS version 25.0 statistical package for MS Windows. Statistical significance was considered for *p* < 0.05.

## 3. Results

The final results study sample consisted of 59 patients (20 patients in groups 1 and 2 and 19 patients in group 3), of which 86.4% were women (*n* = 51) and 13.6% were men (*n* = 8). The mean (standard deviation [SD]) age was 60.7 ± 9.7 years (range 39–82). Table 1 presents the demographic data of the global study sample according to the treatment group. No statistically significant differences were observed in any of the variables between the different treatment groups (Table 1).

In relation to the clinical variables, pain (Table 2) was seen to decrease significantly over time, regardless of the treatment group involved. Specifically, pain intensity decreased significantly in group 1 up until visit four, after which the pain intensity remained without significant changes up until visits five (at one month) and six (at three months) (Table 3). In group 2, the pain intensity likewise decreased significantly until visit four, after which no significant changes were observed with respect to the pain scores recorded at one and three months. Lastly, in group 3, the pain intensity was seen to decrease at visits three and four with respect to baseline, though after one and three months of follow-up, the pain increased again, reaching the baseline values.

After three months, the pain levels in group 1 were significantly lower than those in group 3.

With regard to patient quality of life, the OHIP-14 scores were seen to decrease significantly over time in groups 1 and 2, with no significant changes in group 3 (Table 4).

On examining the anxiety and depression profiles, groups 1 and 2 showed a significant decrease in anxiety in the course of the study, with no significant changes in group 3. Anxiety among the patients in group 3 was significantly greater than in either of the other two treatment groups. In turn, the depression levels were seen to decrease significantly over time, independently of the treatment provided. The interaction between treatment and visit was nonsignificant (Table 5 and Table 6).

On analyzing the severity of the OLP lesions based on the Thongprasom score, significant reductions were recorded at the end of treatment versus baseline in both groups 1 (*p* < 0.001) and group 2 (*p* < 0.011). However, no significant variations were observed in group 3 (*p* = 0.058). There were no significant differences in severity scores between groups 1 and 2 (Figure 2). No adverse effects were recorded during the study in any of the treatment groups.

## 4. Discussion

Photodynamic therapy and photobiomodulation are safe and non-invasive treatment strategies in patients with OLP, with the important advantage of having no major side effects. According to different systematic reviews [8,10], these are attractive therapeutic tools and should be considered for use in patients with OLP.

Oral lichen planus is a chronic mucocutaneous disorder. Topical and systemic corticosteroids have been widely used as the first treatment option for OLP. However, new alternatives are needed for patients with refractory disease or who do not respond to the standard treatments [3].

In 2006, Aghahosseini et al. [18] introduced PDT as an alternative treatment for OLP and recorded a 44.3% decrease in oral OLP lesion size with this technique. In 2019, Lavaee and Shadmanpour [20] reported that PDT could be used as an alternative along with standard treatments or as a new management strategy for refractory OLP.

On the other hand, the beneficial properties of PBM are attributed to its effects upon a range of molecular mechanisms, including an increase in adenosine triphosphate (ATP) levels with the production of reactive oxygen species (ROS) and the consequent activation of transcription factor NF-kB and of survival and immune signaling pathways, as well as the modulation of pro- and anti-inflammatory cytokines, and growth factors [30,31]. These effects are particularly relevant in the context of OLP since the pathogenesis of this disease is characterized by an inflammatory infiltrate with the important participation of T lymphocytes [1,6]. In this regard, PBM has been reported to produce significant reductions in pro-inflammatory cytokines, including TNFα, IFNγ, and IL-1β, with an increase in the release of IL-4, IL-10, and IL-13, as well as of TGFβ [31].

Photodynamic therapy is based on the local or systemic application of a photosensitive compound—the photosensitizer, which is intensely accumulated in pathological tissues. The photosensitizer molecules absorb light of the appropriate wavelength, initiating the activation processes leading to the selective destruction of the inappropriate cells [30,32] Cosgarea et al. [22] showed that following methylene blue-PDT there was a significant decrease in the relative number of CD4+ and CD8+ T-cells in mucosal OLP-lesions. They also demonstrated that activated peripheral CD4 + CD134+ and CD8 + CD137+ T-cells diminished.

The patient groups included in the present study were homogeneous in terms of age and gender distribution, the type of OLP, and the evolution of the lesions, being consistent with previous studies on the presentation of OLP [3,4,5,6]. Some authors have reported that conventional therapy with topical corticosteroids is associated with better control of both OLP pain and lesion size compared with PBM [33]. In contrast, Dillenburg et al. [26] found PBM to offer superior efficacy in controlling the symptoms of OLP. The assessment of analgesic efficacy is central to all studies on the efficacy of treatment for patients with OLP, and the most widely used tool in this respect is the visual analog scale (VAS). An interesting finding in our study was that the use of topical triamcinolone acetonide 0.1% reduced the symptoms during treatment, though subsequently, the pain levels returned to the baseline scores, thereby making continuous use of the medication necessary. In 2017, Akaram et al. [7] conducted a systematic review of the efficacy of laser lower light therapy (LLLT) versus corticosteroid use in patients with OLP and found LLLT to be more effective than the drug treatment in adults. The authors underscored that the scientific evidence is very weak, however.

With regard to the efficacy of PDT versus corticosteroid treatment, data obtained in recent years are inconclusive. Mostafa et al. [12] reported more evident pain control and OLP lesion reduction among the patients treated with PDT than those administered corticosteroids. Bakhitari et al. [14] and Maloth et al. [19] found PDT to be as effective as topical corticosteroid use in treating OLP. However, Saleh et al. [21] reported that corticosteroids are significantly superior to PDT in reducing the symptoms and the number of relapses of the disease. In turn, a meta-analysis carried out in 2020 by investigators at the University of Sichuan (China) [10] described PDT as a second-line treatment option for patients with OLP. In situations where oral or topical corticosteroid use is contraindicated, in patients with resistance to these drugs, or in cases requiring repeated treatment over a short period of time, PDT appears to be a very promising therapeutic option.

Our data are consistent with those published by Ferri et al. [25]. In a randomized, controlled, double-blind trial, these authors applied a larger number of sessions, and the topical corticosteroid used was clobetasol propionate 0.05%. The mentioned study found PDT to be an effective tool and an alternative for patients with symptomatic OLP.

Mirza et al. [13] found that although PBM proved useful in the management of OLP, the efficacy index (i.e., the improvement of symptoms) in the PDT group was significantly better than in either the PBM group (*p* = 0.001) or in the corticosteroid group (*p* = 0.001). These findings are very similar to our own.

It must be noted that we found difficulties in establishing comparisons among PDT, PBM, and topical corticosteroids in the treatment of OLP since the OLP management protocols are very heterogeneous [7,8,9], with variations in terms of the photosensitizers used, the light sources, irradiation dosimetry, the number of treatment sessions and the duration of follow-up and monitoring. Furthermore, the levels of bias are high. These discrepancies between studies point to the need for multicenter and homogeneous trials involving standardized parameters in order to compare PDT, PBM, and topical corticosteroid treatment, to draw firm conclusions.

Although none of the patients in any of the three treatment groups of our study suffered side effects, some investigators [18] reported burning sensations and swelling. Nevertheless, these patient complaints were minor problems compared with the multiple side effects of corticosteroids, such as oral candidiasis, delayed lesion healing, thinning of the mucosa, dry mouth, hyperglycemia, Cushing syndrome, etc. [3,34].

As strong points of the present study, mention must be made of the fact that patient oral quality of life and psychological profile were analyzed because these are important factors in OLP [35,36]. In this respect, the quality of life and anxiety levels were better in groups 1 and 2 than in the corticosteroid treatment group. As limitations of the present study, mention must be made of the use of the Thongprasom score for clinical assessment, as it has been often used in studies on OLP [6,22]. However, this scoring instrument does not take the number of OLP lesions per patient into account.

Both PDT and PBM are practically non-invasive and thus may become treatment alternatives for patients with OLP. A number of advantages warrant a more widespread use of these techniques: they are very effective, exert no toxic effects, allow repeated application, and are not demanding in terms of the material and equipment needed. The feedback provided by the patients is moreover favorable since their quality of life is clearly improved as a result of such treatment. Nevertheless, the diversity of existing PDT protocols and the contradictory data on their efficacy implies that further studies are needed in this field.

## 5. Conclusions

The results obtained indicate that both photodynamic therapy and photobiomodulation once a week for four weeks are safe and non-invasive treatment options, with the important advantage of lacking adverse effects. Overall, the article is interesting for oral medicine practitioners and shows new data on photodynamic therapy and photobiomodulation on OLP. Further studies are needed to confirm it.

## Figures and Tables

**Figure 1 pharmaceutics-15-00030-f001:**
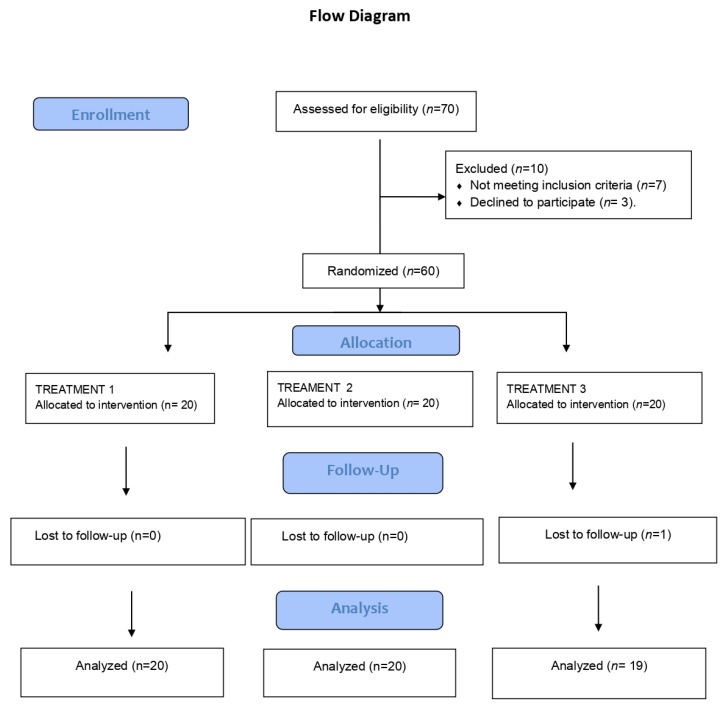
Flow diagram.

**Figure 2 pharmaceutics-15-00030-f002:**
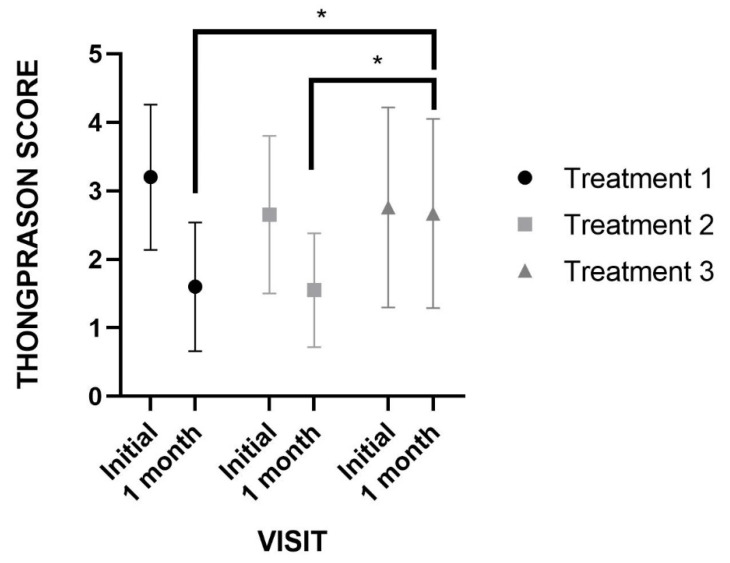
Thongprasom score, change from Baseline-1 month (Score 0–5). * *p* < 0.05.

**Table 1 pharmaceutics-15-00030-t001:** Descriptive and comparative study of the variables between the different treatment groups. No statistically significant differences were observed between the groups for any of the variables.

	Total	Treatment			Test	*p*-Value
		1	2	3		
**Age**	60.7 (9.7)	63.5 (9.5)	61.3 (11.1)	57.2 (7.7)	F(2;56) = 2.16	0.125
**Gender**					χ^2^(2) = 4.10	0.128
Male	8 (13.6)	1 (5.0)	2 (10.0)	5 (26.3)		
Female	51 (86.4)	19 (95.0)	18 (90.0)	14 (73.7)		
**Smoking**					χ^2^(4) = 8.52	0.074
Yes	6 (10.2)	1 (5.0)	1 (5.0)	4 (21.1)		
Ex-smoker	11 (18.6)	7 (35.0)	2 (10.0)	2 (10.5)		
Non-smoker	41 (71.2)	12 (60.0)	17 (85.0)	13 (68.4)		
**Alcohol**					χ^2^(6) = 10.49	0.106
No	37 (62.7)	14 (70.0)	15 (75.0)	8 (42.1)		
1 glass/w	14 (23.7)	5 (25.0)	4 (20.0)	5 (26.3)		
2–3 glasses/w	0 (0.0)	0 (0.0)	0 (0.0)	0 (0.0)		
Social drinker	8 (13.6)	1 (5.0)	1 (5.0)	6 (31.6)		
**Duration of OLP (years)**	4 (3.4)	4.8 (4.3)	3.8 (3.4)	2.5 (1.7)	F(2;56) = 2.26	0.114
**Location lesions**					χ^2^(6) = 5.51	0.48
Others	5 (8.5)	2 (10.0)	3 (15.0)	0 (0.0)		
Buccal mucosa	33 (55.9)	9 (45.0)	10 (50.0)	14 (73.7)		
Gums	15 (25.4)	6 (30.0)	5 (25.0)	4 (21.1)		
Tongue	6 (10.2)	3 (15.0)	2 (10.0)	1 (5.3)		
**CLINICAL LESIONS**					χ^2^(4) = 2.447	0.654
Reticular	25 (42.4)	7 (35.0)	10 (50.0)	8 (42.1)		
Atrophic	20 (33.9)	6 (30.0)	7 (35.0)	7 (36.8)		
Erosive	14 (23.7)	7 (35.0)	3 (15.0)	4 (21.1)		

**Table 2 pharmaceutics-15-00030-t002:** Mean values (standard deviation SD) and statistical contrasts between pain treatments. Pain numerical rating scale score (Scale: 0–10).

	Visit, *Mean (SD)*						Within-Subject Effects ^†^	
	1	2	3	4	5	6	Visit	Treatment * Visit
							*F*(df); *p*-Value (*η*^2^)	*F*(df); *p*-Value (*η*^2^)
**PAIN SCALE**							*F*(5;275) = 46.14; ***p* < 0.001** (0.456)	*F*(10;275) = 5.80; ***p* < 0.001** (0.174)
Treat. 1	6.70 (3.44)	5.65 (3.07)	4.30 (2.83)	2.80 (1.94)	1.95 (1.79)	2.05 (2.28)		
Treat. 2	6.50 (2.86)	5.50 (2.72)	4.25 (2.71)	3.25 (2.57)	3.10 (2.17)	3.20 (2.38)		
Treat. 3	4.94 (3.02)	4.17 (2.50)	3.33 (2.06)	2.83 (2.12)	3.61 (2.93)	4.06 (2.98)		
*Total*	6.09 (3.16)	5.14 (2.81)	3.98 (2.57)	2.97 (2.20)	2.86 (2.39)	3.07 (2.64)		

**^†^** Sphericity assumed. df: degrees of freedom. *η*^2^: partial eta-square (effect size).

**Table 3 pharmaceutics-15-00030-t003:** Differences in means and *p*-values between the treatment groups at each of the visits.

Visit	Difference in Means (*p*-Value)		
	Treatment 1	Treatment 2	Treatment 3
1-2	1.05 (0.006)	1.00 (0.011)	0.77 (0.162)
1-3	2.40 (<0.001)	2.25 (<0.001)	1.61 (0.029)
1-4	3.90 (<0.001)	3.25 (<0.001)	2.11 (0.001)
1-5	4.75 (<0.001)	3.40 (<0.001)	1.33 (0.272)
1-6	4.65 (<0.001)	3.30 (<0.001)	0.88 (1)
2-3	1.35 (0.012)	1.25 (0.026)	0.83 (0.632)
2-4	2.85 (<0.001)	2.25 (<0.001)	1.33 (0.095)
2-5	3.70 (<0.001)	2.40 (0.001)	0.55 (1)
2-6	3.60 (<0.001)	2.30 (0.009)	0.11 (1)
3-4	1.50 (<0.001)	1.00 (0.044)	0.5 (1)
3-5	2.35 (<0.001)	1.15 (0.047)	−0.2 (1)
3-6	2.25 (0.001)	1.05 (0.048)	−0.7 (1)
4-5	0.85 (0.143)	0.15 (1)	−0.7 (0.352)
4-6	0.75 (1)	0.05 (1)	−1.2 (0.223)
5-6	−0.1 (1)	−0.1 (1)	−0.4 (1)

**Table 4 pharmaceutics-15-00030-t004:** Mean values (standard deviation SD) and statistical contrasts between treatments of the OHIP-14; anxiety and depression.

	Visit, *Mean (SD)*			Within-Subject Effects ^†^	
	Initial (1)	1 Month (2)	3 Months (3)	Visit	Treatment * visit
				*F*(df); *p*-Value (eta2)	*F*(df); *p*-Value (eta2)
**OHIP-14**				*F*(2;110) = 46.20; ***p* < 0.001** (0.457)	*F*(4;110) = 2.53; ***p* = 0.046** (0.088)
Treat. 1	18.05 (9.43)	12.75 (7.17)	9.10 (5.21)		
Treat. 2	22.55 (9.12)	17.50 (9.22)	14.30 (7.95)		
Treat. 3	25.94 (12.35)	23.94 (12.26)	22.06 (12.15)		
*Total*	22.05 (10.64)	17.86 (10.54)	14.91 (10.11)		
**Anxiety**				*F*(2;110) = 25.75; ***p* < 0.001** (0.319)	*F*(4;110) = 4.39; ***p* = 0.002** (0.138)
Treat. 1	12.35 (3.79)	9.05 (5.28)	7.20 (4.23)		
Treat. 2	12.55 (3.53)	9.05 (2.56)	6.60 (4.37)		
Treat. 3	10.94 (3.40)	11.06 (3.64)	10.17 (3.37)		
*Total*	11.98 (3.59)	9.67 (4.04)	7.91 (4.25)		
**Depression**				*F*(2;110) = 9.74; ***p* < 0.001** (0.151)	*F*(4;110) = 0.67; *p* = 0.615 (0.024)
Treat. 1	4.85 (5.06)	4.55 (4.92)	4.25 (4.44)		
Treat. 2	4.00 (2.36)	3.70 (2.36)	3.40 (2.23)		
Treat. 3	6.00 (4.02)	5.87 (3.43)	5.33 (2.98)		
*Total*	4.91 (3.98)	4.71 (3.75)	4.33 (3.35)		

**^†^** Sphericity assumed. df: degrees of freedom. *η*^2^: partial eta-square (effect size).

**Table 5 pharmaceutics-15-00030-t005:** Differences in means and *p*-values between the treatment groups at each of the visits.

Follow-Up	Difference in Means (*p*-Value)		
	Treat. 1	Treat. 2	Treat. 3
OHIP-14 Initial-1 month	5.30 (<0.001)	5.05 (<0.001)	2 (0.363)
Initial-3 months	8.95 (<0.001)	8.25 (<0.001)	3.88 (0.065)
1 month-3 months	3.65 (0.003)	3.20 (0.01)	1.88 (0.27)
**Anxiety** Initial-1 month	3.30 (0.001)	3.50 (<0.001)	−0.11 (1)
Initial-3 months	5.15 (<0.001)	5.95 (<0.001)	0.77 (1)
1 month-3 months	1.85 (0.044)	2.45 (0.029)	0.88 (1)
**Depression** Initial-1 month	0.3 (0.449)	0.3 (0.449)	0.13 (0.388)
Initial-3 months	0.6 (0.385)	0.6 (0.385)	0.67 (0.119)
1 month-3 months	0.3 (0.95)	0.3 (0.95)	0.54 (0.303)

**Table 6 pharmaceutics-15-00030-t006:** Differences in means and *p*-values between the treatment groups at each of the visits.

Visit	Difference in Means (*p*-Value)		
	Treat. 1 vs. Treat. 2	Treat. 1 vs. Treat. 3	Treat. 2 vs. Treat. 3
**OHIP-14** Initial	−4.50 (0.521)	7.89 (0.066)	−3.39 (0.948)
1 month	−4.75 (0.379)	−11.19 (0.002)	−6.44 (0.135)
3 Months	−5.20 (0.198)	−12.96 (<0.001)	−7.76 (0.026)
**Anxiety** Initial	−0.20 (1)	1.41 (0.698)	1.61 (0.521)
1 month	0.00 (1)	−2.01 (0.385)	−2.01 (0.385)
3 months	0.60 (1)	−2.97 (0.083)	−3.57 (0.026)
**Depression** Initial	0.85 (1)	−1.15 (1)	−2.00 (0.38)
1 month	0.85 (1)	−1.12 (1)	−1.97 (0.331)
3 months	0.85 (1)	−0.58 (1)	−1.43 (0.582)

## Data Availability

The datasets generated and/or analyzed during the current study are not publicly available due contains sensitive data that can identify but are available from the corresponding author on reasonable request.

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
