# Peer review of "Photodynamic Therapy, Photobiomodulation and Acetonide Triamcinolone 0.1% in the Treatment of Oral Lichen Planus: A Randomized Clinical Trial"

_pharmaceutics, 2022, doi:10.3390/pharmaceutics15010030_

Round 1

Reviewer 1 Report

The article titled ‘Photodynamic Therapy, Photobiomodulation and Acetonide Triamcinolone 0.1% in The Treatment of Oral Lichen Planus: A randomized clinical trial’ may be an useful contribution to the journal; the research is sound, seems well conducted and the statistical apparatus employed is also appropriate, considering sample size and chosen tests; however, few changes should be taken into consideration:

Line 178- authors state ‘statistical contrasts between pain treatments’ –should be rephrased as the treatments are not analgesic treatments, and the reader would otherwise be confused.

Authoers should mention whether time*treatment interactions were found in repeated measures ANOVA across time/various groups. Charts/graphs would benefit the article.

Also, at least data depicted in Table 7 Thongprasom score results should be presented as a chart, in the interest of the reader.

Authors should take into consideration and elaborate on the various reactions (e.g. pain) associated to the treatment sessions, in this trial or in other articles in medical literature in the filed; authors should explain or rephrase accordingly the phrase in the conclusion of both the article and the abstract regarding total lack of adverse effects, in this regard.

Line 86- authors are advised to check the link in the text.

Grammar and punctuation must also be carefully checked within the entire article  (e.g. lines 22-23 –missing parantheses, as well as line 116 etc).

Author Response

The article titled ‘Photodynamic Therapy, Photobiomodulation and Acetonide Triamcinolone 0.1% in The Treatment of Oral Lichen Planus: A randomized clinical trial’ may be an useful contribution to the journal; the research is sound, seems well conducted and the statistical apparatus employed is also appropriate, considering sample size and chosen tests; however, few changes should be taken into consideration:

Thank you for the positive comments and suggestions

Line 178- authors state ‘statistical contrasts between pain treatments’ –should be rephrased as the treatments are not analgesic treatments, and the reader would otherwise be confused.

It has been changed

Authoers should mention whether time*treatment interactions were found in repeated measures ANOVA across time/various groups. Charts/graphs would benefit the article.

Also, at least data depicted in Table 7 Thongprasom score results should be presented as a chart, in the interest of the reader.

It has been added

Authors should take into consideration and elaborate on the various reactions (e.g. pain) associated to the treatment sessions, in this trial or in other articles in medical literature in the filed; authors should explain or rephrase accordingly the phrase in the conclusion of both the article and the abstract regarding total lack of adverse effects, in this regard.

It has been modified

Line 86- authors are advised to check the link in the text.

Grammar and punctuation must also be carefully checked within the entire article  (e.g. lines 22-23 –missing parantheses, as well as line 116 etc).

It has been checked

Reviewer 2 Report

Dear Authors,

The article entitled "Photodynamic Therapy, Photobiomodulation and Acetonide Triamcinolone 0.1% in The Treatment of Oral Lichen Planus: A randomized clinical trial"comprises a study evaluating the efficiency of photodynamic therapy and photobiomodulation added to topical corticosteroid treatments in OLP patients.

Table 1. In Line 166 add which statistical test was used. Regarding the location of the lesions, the Cheek mucosa should be replaced with buccal mucosa. Also replace SIGNS with clinical lesions.

Line 291: the authors mean the  great majority of patients with OLP present multiple clinical types of lesions. A patient cannot have multiple clinical forms of the disease.

The patients groups were not homogeneous in terms of the duration of the lesions. 4.8 years of OLP lesions evolution is different from 2.5 years of evolution. And this is a weak point of the study. Can the authors mention if these patients were previously treated for OLP? Or all were new patients?  Previous treatment can influence not only the lesions but also the OHIP scores and the patient's perception of OLP.

Line 299: In the conclusions should be added that photodynamic therapy and photobiomodulation bring some (slight?)improvement in OLP lesions.

Overall the article is interesting for oral medicine practitioners and shows new data on photodynamic therapy, and photobiomodulation on OLP. Further studies are needed to confirm it.

I recommend a minor revision.

Best regards!

Author Response

The article entitled "Photodynamic Therapy, Photobiomodulation and Acetonide Triamcinolone 0.1% in The Treatment of Oral Lichen Planus: A randomized clinical trial"comprises a study evaluating the efficiency of photodynamic therapy and photobiomodulation added to topical corticosteroid treatments in OLP patients.

Thank you for the positive comments and suggestions

I carefully read the manuscript and commented my considerations for the study. I hope it would be a help in improving your manuscript

Table 1. In Line 166 add which statistical test was used. Regarding the location of the lesions, the Cheek mucosa should be replaced with buccal mucosa. Also replace SIGNS with clinical lesions.

It has been replaced

Line 291: the authors mean the  great majority of patients with OLP present multiple clinical types of lesions. A patient cannot have multiple clinical forms of the disease.  I agree

The patients groups were not homogeneous in terms of the duration of the lesions. 4.8 years of OLP lesions evolution is different from 2.5 years of evolution. And this is a weak point of the study. Can the authors mention if these patients were previously treated for OLP? Or all were new patients?  Previous treatment can influence not only the lesions but also the OHIP scores and the patient's perception of OLP.

I agree that the time of evolution may have an influence

 All patients in all groups were included in the same way.. Subjects under 18 years of age were excluded from the study, as were patients subjected to corticosteroid therapy in the two months prior to the study

Line 299: In the conclusions should be added that photodynamic therapy and photobiomodulation bring some (slight?)improvement in OLP lesions.

Overall the article is interesting for oral medicine practitioners and shows new data on photodynamic therapy, and photobiomodulation on OLP. Further studies are needed to confirm it.

Ok

Round 2

Reviewer 1 Report

Manuscript has been significantly improved.